# Interaction Mechanism between Inter-Organizational Relationship Cognition and Engineering Project Value Added from the Perspective of Dynamic Impact

**Mengyu Xu [1], Xun Liu [2,*], Zhen Bian [2] and Yufan Wang [3]**

[1] School of Business, Suzhou University of Science and Technology, Suzhou 215000, China; 2211051018@post.usts.edu.cn
[2] School of Civil Engineering, Suzhou University of Science and Technology, Suzhou 215000, China; 2313031146@post.usts.edu.cn
[3] School of International Education, Suzhou University of Science and Technology, Suzhou 215000, China; wangyufan1009@163.com
[*] Correspondence: liuxun8127@usts.edu.cn; Tel.: +86-13675102267

**Abstract:** Projects involve inter-organizational relationship cognition, which is central to collaborative engineering project value added. Interest in value added in the project lifecycle is mounting and gaining increasing attention in the research literature. However, little is known about how inter-organizational relationship cognition facilitates value added and how such cognition pushes a project toward higher end-states of value. The existing literature mainly analyzes and studies value added on functional analysis and cost control. There are predominantly static analyses of the factors that influence value added in studies. The guiding role of value added has not been adequately explored in the studies on the influencing factors of value added. Utilizing a combination of Structural Equation Modeling (SEM) and Fuzzy Cognitive Maps (FCMs), this study addresses how inter-organizational relationship cognition influences engineering project value added, identifying complex structures of interaction and cognition dynamics. Results indicate that: (1) A hybrid SEM–FCM method can be able to model dynamic interactions between inter-organizational relationship cognition and value added; (2) trust and shared vision have positive effects on in-role behavior and extra-role behavior. Shared vision has a negative effect on opportunistic behavior. In-role behavior and extra-role behavior have a positive impact on value added, while opportunistic behavior has a negative impact. Organizational behavior is an important mediating variable to explain the interaction between inter-organizational relationship cognitions and value added. This hybrid method explores the potential mechanisms of inter-organizational relationship cognition on project value added from novel perspectives on construction project management practices, proposing practical advice for further project management.

**Keywords:** engineering project value added; inter-organizational relationship cognition; organizational behavior; structural equation modeling; fuzzy cognitive map



## 1. Introduction

As the external environment becomes increasingly dynamic and complex and the pace of technological updates accelerates, the traditional criteria for project success, which focus on achieving the three main goals of 'cost, time and quality', no longer align with the patterns of societal development [1]. Many experts agree that value delivery has now become the standard for measuring the success of engineering projects [2–4]. Simultaneously, the advent of advanced technologies has profoundly impacted engineering projects, enhancing not only the efficiency of project delivery but also transforming the value these projects create [5]. This shift has driven a gradual change in project management, emphasizing the importance of responding to technological innovations to capture additional strategic

value. Scholars are increasingly focusing on the actual value added to a project, rather than merely the outputs delivered, as a measure of project success [6].

The concept of value added originates from value chain theory, first proposed by Porter [7]. It refers to the creation of new value that leads to an increase in the existing value and is widely applied in the field of engineering. Engineering projects not only provide profit opportunities for construction enterprises but also serve as important platforms for implementing value chain management. It emphasizes not only achieving the basic objectives outlined in contracts but also long-term goals such as better communication, fewer disputes, and the involvement of new technologies to enhance the project's sustainability. Analyzing project management processes through the lens of value chain and value theory helps shift the emphasis from a 'task-oriented' to a 'value-oriented' approach. This enables project participants to engage in targeted value-added activities, achieving the contractual goals while also creating greater economic and social benefits, continuously enhancing their competitive advantages for construction enterprises. Thus, to overcome the plight of high cost and low efficiency for engineering projects, it is essential to probe the interaction between influencing factors and value added.

Owing to the nature of engineering projects, countervailing forces are created to detect and improve the value added: (1) Engineering projects involve numerous stakeholders, including owners, government bodies, contractors, suppliers, and the community. Conflicts of interest may arise among stakeholders, and balancing these interests is a major challenge for value added [8,9]; (2) during the project process, participants may be unwilling or unable to fully share key information, leading to a lack of transparency. This increases the project's uncertainty and risk, causes information asymmetry, and affects the accuracy and timeliness of decision-making [10,11]; (3) during contract execution, disputes over responsibilities and rights may arise, increasing the complexity of project management. Participants may face issues such as poor performance and delays, which can affect the smooth progress of the project and the realization of its value [12,13]. The need to view the value added as a project management requirement is a relatively new concept and topic of discussion that requires moving away from standard project delivery tools and techniques associated with hard paradigms and focusing on the social structures and interactions that occur in project delivery. The key is to satisfy the value quest and inherent expectations of the various participants, which is based on a clear understanding of the inter-organizational relationship cognition among the project participants [14]. Inter-organizational relationship cognition refers to the understanding and cognition of the relationship between the various participants in the project [15]. Based on a review of the existing research, some researchers have explored how inter-organizational relationship cognition can promote the value added in engineering projects. Faems studied the link between relational governance methods and value realization in asymmetric new venture development alliances [16]. Bossink argued that the key to project value added lies in the efficient acquisition and timely processing of knowledge and that collaborative innovation facilitates the flow of knowledge, thereby contributing to the increase in project value [17]. These studies fail to fully capture the complex interplay between inter-organizational relationship cognition and value added, often simplifying the actual interaction processes. It often overlooks this dynamic nature, limiting our understanding of how inter-organizational relationship cognition influences value added throughout the project lifecycle.

Moreover, previous studies on the value added in engineering projects are very limited, most of them are only functional analyses and cost controls, such as through innovative financing methods (such as a public–private partnership model—PPP); efficient project management and risk management strategies to realize the value added of projects [18–20]; or analyses of how to add economic and social value to the project by meeting green building standards (such as energy saving, reducing carbon emissions, improving indoor environmental quality, etc.) [21,22]. The value added dynamically evolves as the project progresses and external conditions change. Meanwhile, the influencing factors dynamically change over time, interact with each other, and are interdependent rather

than isolated. A static analysis of factors alone makes it difficult to formulate effective decisions to realize the value added. As the value added develops, it is crucial to examine the dynamic interplay between inter-organizational relationship cognition and the value added in project management. The components of engineering projects are highly interconnected, with changes in one influencing factor often triggering systemic shifts in the value added. As a result, neither theoretical knowledge nor practical experience alone can effectively guide the value added in engineering projects. Therefore, it is essential to examine how cognition factors impact performance within these dynamic interactions. We propose a conceptual model of the cognition–value relationship that incorporates trust and shared vision, drawing from strategy literature and a dynamic capabilities approach. This model aims to enhance our understanding of how inter-organizational relationship cognition influences the value added. By integrating cognition and value into a cohesive framework, this model could help address potential challenges faced by project managers and practitioners. In addition, project delivery is essentially a social process, fundamentally based on a networked organizational form that creates value through the mutual cooperation of participants. The behavior of project participants is also closely related to the engineering project value added [23]. The realization of value added relies on the understanding of inter-organizational relationship cognition and the coordination of organizational behavior. Therefore, introducing organizational behavior as a mediating variable to study the mechanism by which the understanding of inter-organizational relationship cognition affects the value added is essential. In this context, this paper specifically argues the impact of different dimensions of inter-organizational relationship cognition (trust, shared vision) on the value added and on three different types of organizational behavior (in-role behavior, extra-role behavior, and opportunistic behavior) as a behavioral outcome. It elucidates how inter-organizational relationship cognition can enhance the value added and also seeks to clarify the dynamic evolution mechanism of the value added to elevate the overall value-added level of engineering projects achieved by combining SEM and FCM. SEM is a multivariate statistical analysis method commonly used for linear regression and causal relationship verification [24]. However, there is a lack of literature on the dynamic nature of value-added evolution. Managing the value added effectively is challenging when individual factors are treated in isolation. Therefore, modeling and evaluating value-added processes and their impact on dynamic evolution requires a robust approach. The FCM approach enables systematic causal propagation and supports 'what-if' scenarios for modeling complex systems [25]. FCM has been widely applied across various fields, including decision-making, risk analysis, and knowledge management [26]. However, traditional FCMs have certain limitations: (1) they may not accurately capture the system's characteristics, as the factors of concept nodes are often not verified, and (2) determining the weights between nodes through expert knowledge acquisition can be time-consuming [27]. To address these challenges, path coefficients derived from SEM can be utilized to construct the FCM model, reducing the potential for errors and biases in expert-driven FCM construction.

This study aims to achieve the following objectives: (1) to examine which inter-organizational relationship cognition has an impact on the value added and how it affects the value added; (2) to use SEM and FCM to study the dynamic interaction between inter-organizational relationship cognition and the value added, which compensates for the shortcomings of static SEM research and resolves the subjectivity and ambiguity problems caused by expert knowledge; (3) to explore the mediating role of organizational behavior in the relationship between inter-organizational relationship cognition and the value added. In the remainder of the paper, the following structure is followed: Section 2 defines inter-organizational relationship cognition and organizational behavior and reviews related studies on value added. Section 3 demonstrates survey instruments and the validity of the questionnaire data. Section 4 presents the research methods with a hybrid SEM–FCM analysis. It extends an SEM model that uncovers the relationship between inter-organizational relationship cognition, organizational behavior, and value added and presents the FCM

model analyses, including predictive and diagnostic analyses and hybrid analysis. Section 5 presents the discussions, followed by the conclusions in Section 6.

## 2. Literature Background and Model Establishment

### 2.1. Inter-Organizational Relationship Cognition

Lau and Rowlinson emphasized that successful project management relies on balancing formal and informal structures [28]. This balance is built upon a well-defined agreement or framework of working relationships and may also encompass a socially based inter-organizational relationship cognition. It refers to participants' understanding, perception, and evaluation of the relationship between their organization and external organizations [15]. This cognition involves participants' awareness of aspects such as collaboration, competition, trust, information exchange, and shared goals between organizations. It is difficult to trace and describe due to its abstract form and multi-faceted nature. What we are aware of is that it brings project participants together and encourages commitment to the project. Hence, it requires behavioral outcome, that is, trust and shared vision, to qualify it [28]. Due to its inherent temporality and complexity, engineering projects bring great uncertainty to project implementation and practice [14]. Inter-organizational relationship cognition significantly impacts engineering project management by fostering cooperation, trust, and efficient communication among project stakeholders. Positive cognition ensures that strategic goals are aligned, risks are managed effectively, and overall project success is achieved, ultimately contributing to the timely and cost-effective completion of engineering projects. Kauppila explored how social cognition influences employees' information exchange and the formation of perceptions in inter-organizational collaborations [29]. Trust is defined as the participants in the project believing that the partner will not take advantage of the project's vulnerability to damage the project interests, will comply with the contract, and will complete their tasks [30]. According to Rowlinson and Steve's research, building trust between project parties helped align relationships, promoted ongoing collaboration, and reduced regulatory and communication costs [31]. This simplifies the contract and regulation development process, improves management efficiency, reduces construction time, and ensures project quality. Jiang and Zhao believed that trust is the cornerstone of fruitful cooperation and alliances, which can promote the consolidation of relationships between partners [32]. Shared vision refers to a common understanding and agreement among members of an organization or project team about future goals and decisions [33,34]. Chi explored the significant impact of shared vision on value creation in large-scale projects and conducted a multi-set analysis between clients and prime contractors [35]. Suprapto argued that parties with a shared vision are more likely to accept their own responsibilities, identify their own specific goals, and test hypothetical models by using partial least squares structural equation models [36]. In engineering projects, the cooperation between the participants is not only a mechanical task division but also involves a complex process of interpersonal relationships, communication, and decision-making. Therefore, this paper focuses on inter-organizational relationship cognition and analyzes how it affects organizational behavior to generate engineering project value added. Studying the engineering project value added from the perspective of project participants' inter-organizational relationship cognition helps to improve team synergy, reduce internal friction, stimulate employee potential, increase adaptability, and enhance trust and cooperation. This approach promotes information sharing and resource integration, creates a harmonious work environment, and fosters creativity and innovation. It enables the team to better respond to external changes and uncertainties, thereby improving project success rates and overall value from multiple dimensions.

### 2.2. Organizational Behavior

Organizational behavior is an important intra-organizational phenomenon which refers to the response of an individual, group, or organization to exogenous and endogenous stimuli in the context of an organization [37,38], divided into cooperative behavior and opportunistic behavior [39,40]. Organizational behavior, as the behavior patterns and interaction modes of project participants within the organization, directly affects the execution and final outcomes of projects. This helps in understanding their adaptive behaviors following inter-organizational relationship cognition. Such a comprehensive focus not only contributes to the short-term success of projects but also provides a solid foundation for the organization's continuous development and strategic implementation. Villena explored cooperative and opportunistic behaviors in buyer–supplier relationships, analyzing how relationship management can mitigate the negative effects of opportunistic behavior [41]. Wang studied contractual and relational governance, analyzing their combined effects on controlling opportunistic behavior and exploring how cooperative behavior can enhance governance effectiveness [42]. Cooperative behavior refers to promoting the sharing of project goals among all participants. It is a continuous and dynamic process that emphasizes the importance of interaction and communication quality between the parties involved [43]. Tabassi found that the contradictions and conflicts among the participants in the project would intensify their mutual defense, while close cooperation could reduce this phenomenon and contribute to the progress of the project [44]. Their research highlighted the positive role of cooperative behavior in project conflict resolution. Anvuur and Kumaraswamy provided an in-depth study of collaborative behavior in construction projects in terms of in-role behavior and extra-role behavior, providing a multi-angle vision for a more comprehensive understanding of the cooperative behavior of various participants in the project [43]. In-role behavior refers to the mandatory actions taken by all participants to fulfill project responsibilities and adhere to the stipulations outlined in formal documents such as contracts and agreements [45]. These behaviors are based on necessary regulations and constitute the work standards that individuals are expected to follow. They typically include complying with rules and regulations, completing assigned tasks, and actively participating in team collaboration. Extra-role behavior refers to actions that go beyond the scope of work duties and provide additional value to the organization [46]. These behaviors are usually voluntary and spontaneous, and they may be beneficial responses to projects that exceed the standard job requirements. Opportunistic behavior refers to improper actions taken by individuals within an organization or transactional relationship to seek personal gain [47]. These actions include, but are not limited to, providing incomplete or distorted information, breaching contract terms, shirking responsibilities, exploiting the other party's unfavorable situation to exert pressure, and failing to fulfill relational commitments or obligations [48]. Cheung believed that contractors may seek to benefit themselves by concealing their qualifications or bidding at artificially low prices, thereby jeopardizing the quality and progress of the project [49]. Lu's empirical research demonstrates that opportunistic behavior reduces relationship satisfaction among project participants, consequently harming the project's performance [50]. Understanding the organizational behavior of project participants not only helps improve their work performance and satisfaction but also aids the organization in responding to changes and challenges, thereby maintaining a competitive edge. Through effective management of employee behavior, organizations can create a positive work environment that fosters the holistic development of employees and ensures the organization's ongoing success. By promoting cooperative behaviors, reducing opportunistic behaviors, enhancing adaptability, and stimulating employee potential, project success rates and overall value can be elevated on multiple levels, thereby aiding the organization in achieving its long-term strategic goals.

### 2.3. Value Added in Engineering Projects

Value added refers to increasing the net present value of engineering projects and reducing costs by eliminating activities that do not contribute to value creation [51]. This ap-

proach reduces capital investment while also shortening the project duration. Value added in long-term projects implies the fulfillment of latent demand targets that cannot be explicitly agreed upon in the contract but benefit future operations [52,53]. As Wu argued, value addition to engineering projects is more than just a reflection of the project body; it is also an indication of communication, information sharing, trust building, core competencies, and the potential for future collaboration between the parties involved [54]. Browning developed an integrated framework using the key attributes of stakeholder concerns, namely risks and opportunities [55]. Basole argued that the structure of organizational partner networks impacts the creation of greater value within organizations [56]. Additionally, many researchers have studied how to achieve value addition through inter-organizational relationships. Pargar explored the link between relational governance methods and inter-firm value added [57]. Fuentes pointed out that value outcomes can be better achieved through joint efforts between customers and suppliers [58]. Furthermore, researchers have also considered risk management, relational governance, and execution processes to effectively add more value to projects [59,60]. In traditional construction models, the 'output-oriented' approach focuses on accomplishing the specified tasks of a project and delivering the final product. This narrow goal orientation often leads project managers to concentrate on meeting targets related to time, budget, and quality, making them less adaptable to changes in the external environment. In today's rapidly changing social and technological landscape, inflexible project management may struggle to adapt to external changes, increasing project uncertainty. Meanwhile, the continuous emergence of new technologies profoundly impacts engineering projects, not only enhancing project delivery efficiency but also transforming the value that projects can create. This drives the present study to focus on value added in engineering projects, aiming to achieve greater strategic value and technological innovation.

*2.4. Model Establishment*

With cooperative behavior and opportunistic behavior serving as the mediating factor, this study investigates the mechanisms by which trust and shared vision shape engineering project value added. A clear understanding of inter-organizational relationship cognition is very important to realize engineering project value added. If the establishment of inter-organizational trust and the clarity of shared vision cannot be guaranteed, the organizations will easily have different opinions on project-related issues, lack of coordination, and cannot avoid contradictions and conflicts. In addition, organizational behavior plays an important role in completing the construction task and promoting the success of the project and is closely related to the inter-organizational relationship cognition, so it is necessary to explore the static and dynamic mechanism of the action. This paper focuses on the research on the method of realizing the engineering project value added, aiming to explore the effect of inter-organizational relationship cognition on the engineering project value added through organizational behavior. Both conceptually and methodologically, using inter-organizational relationship cognition and organizational behavior as predictors of value added is well-supported. However, there is still a limited understanding of which factors most effectively predict the value added. What sets our model apart is that it (a) introduces the concept of an overall value-added dimension, measuring the direct impact of inter-organizational relationship cognition on value added, and (b) considers three distinct levels of organizational behavior. A hybrid SEM–FCM approach is applied in this study to link inter-organizational relationship cognition and value added to probe the evolutional dynamic relationship. Value-added evaluation can be viewed from a new perspective with the hybrid model in this study. On the basis of the above research and theories, this paper examines the impact of inter-organizational relationship cognition on value added using organizational behavior as the intermediary variable. These hypotheses are developed in order to determine the relationship between them, as shown in the table below:

**H1.** *Trust has a positive correlation with in-role behavior (a), extra-role behavior (b), and has a negative correlation with opportunistic behavior (c).*

**H2.** *Shared vision has a positive correlation with in-role behavior (a), extra-role behavior (b), and has a negative correlation with opportunistic behavior (c).*

**H3.** *In-role behavior (a) and extra-role behavior (b) have a positive correlation with value added; opportunistic behavior (c) has a negative correlation with value added.*

**H4.** *In-role behavior (a), extra-role behavior (b), and opportunistic behavior (c) mediate between trust and value added.*

**H5.** *In-role behavior (a), extra-role behavior (b), and opportunistic behavior (c) mediate between shared vision and value added.*

These hypotheses form a cognition–behavior–value chain model. In this context, we have examined the effects of inter-organizational relationship cognition on value added, extending this chain by incorporating various dimensions of inter-organizational relationship cognition and three distinct types of organizational behavior. Accordingly, these hypotheses support the initial connections within this chain. Taking into consideration the above hypotheses, the SEM model was constructed as shown in Figure 1 in this study.

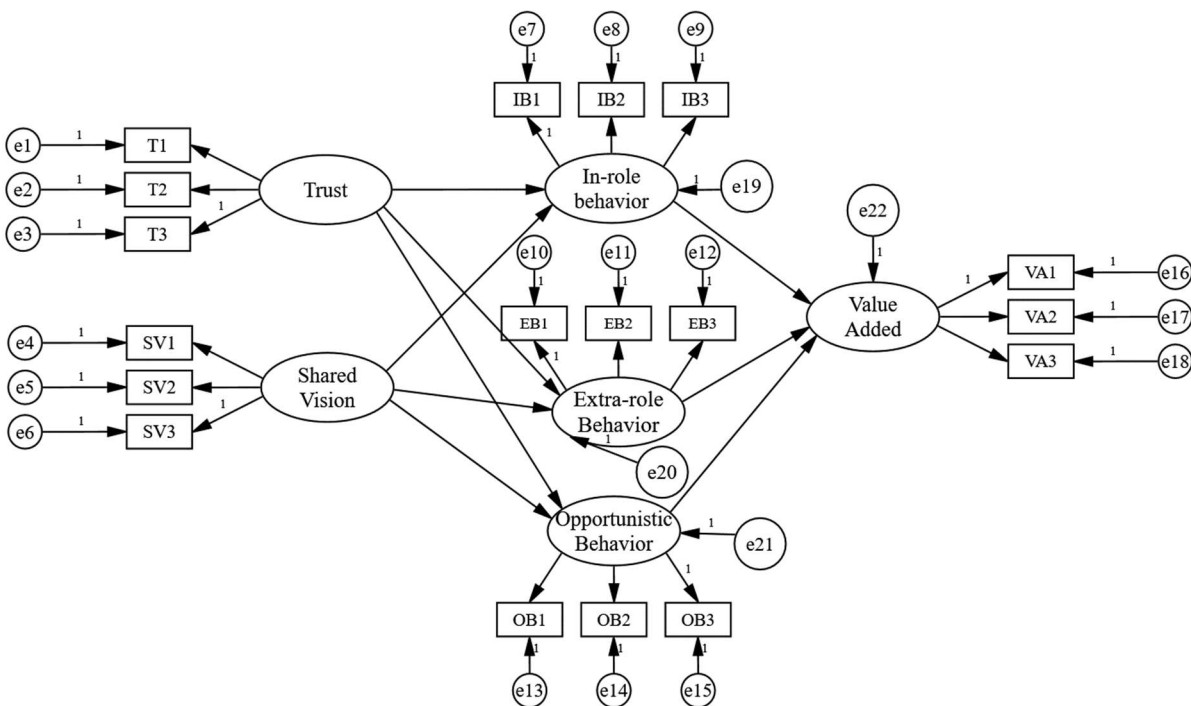

**Figure 1.** SEM model of inter-organizational relationship cognition, organizational behavior, and value added.

### 3. Methodology

*3.1. Survey Instruments*

An assessment of the literature was conducted to determine what measurement methods would ensure the high reliability and validity of the questionnaire items. These items were adjusted to account for the settings of projects, resulting in the creation of the preliminary questionnaire. The assessment of inter-organizational relationship cognition focuses on trust and shared vision. Based on the research by Zhang and Li, three items were used to measure participants' goodwill-based trust, with appropriate wording modifications to

fit the project settings [61,62]. Three items from Lee and Chi's research were adapted to measure shared vision in projects [35,63]. Three items for value added were adapted based on Liu' research, measuring the value added by joint efforts within the organization [52]. According to Zhang's research, six items measure cooperative behavior, divided into in-role behavior and extra-role behavior [64]. The in-role behavior was measured by the extent to which participants fulfilled project responsibilities and complied with explicit provisions, such as contract agreements, including compliance with rules and regulations, completion of work tasks, and active participation in teamwork. The extra-role behavior was measured by the extent that participants go beyond the job description and provide additional value to the organization. The measure of opportunistic behavior covers distorting information and breaking public or implied promises using three items refined according to Lu's research [65]. The questionnaire consists of two parts: A brief description of participants' backgrounds, including the types of projects, their positions, and their work experiences, is presented in the first part of the questionnaire. In the second part, participants were scored based on inter-organizational relationship cognition, organizational behavior, and value added using a 7-point Likert scale. Given the negative nature of opportunistic behavior, several measures were implemented to assess and mitigate potential issues related to social desirability bias, which could compromise the accuracy of behavioral research. Firstly, indirect questioning and randomized response techniques were employed. By mixing sensitive and non-sensitive items, these methods help distract participants and protect their privacy, encouraging more truthful responses. Additionally, the tactic of other-rating was utilized, where some participants were asked to rate their partners' behaviors. Furthermore, participants were assured of anonymity and confidentiality. It was clearly stated at the beginning of the questionnaire that all responses would be kept confidential. Participants' identity information and their answers were stored separately, with anonymous coding used to link the two, ensuring data privacy and preventing any potential leakage.

### 3.2. Sampling and Data Collection

The targeted participants were professional team members familiar with project practices. Before initiating the survey, the data collection tool was piloted to confirm its accuracy and adequacy. Further refinement of the project was achieved through semi-structured interviews. To ensure the survey items aligned with project practices, ten experts from academia and industry were consulted. The interview process was divided into two stages, with the first stage focusing on adapting the items on value added based on the experts' insights. Next, the experts provided their professional opinions on the suitability of the measurements for assessing inter-organizational relationship cognition, organizational behavior, and value added. Based on their feedback, further adjustments were made. Questions were removed or improved in response to the experts' input. For instance, to enhance participants' understanding, some experts recommended describing specific examples of organizational behavior at the beginning of the second part of the questionnaire. Throughout the two-month data collection period, 276 questionnaires were collected. As a result of removing invalid questionnaires with duplicate or missing data, 79.35% of the remaining 219 valid questionnaires were valid. Potential participants were all experienced practitioners in construction projects, including owners, contractors, engineering consultants, and other related organizations. Participants were asked to recall a recent project they had worked on and evaluate the degree of their inter-organizational relationship cognition, organizational behavior, and value added. Table 1 illustrates details about the demographic information and project types of participants. Participants covered all types of engineering projects. This ensures randomness of the sample and reduces interference from specific item characteristics. Meanwhile, 17.4% of participants were owners, 51.6% were contractors, 23.3% were from consultancy, and 7.8% were from other organizations, making the survey more representative. Among the participants, there is a certain percentage of middle and senior managers, and there is no shortage of senior managers, which means that the respondents have a good understanding of the project. In total, 92.7% of participants had a

bachelor's degree or higher, indicating that most participants had a solid knowledge base in their field and were able to make strong judgments. Given that nonresponse bias is a common issue with online surveys, it was specifically tested in this study. As shown in Table 1, a total of 128 participants (58.4%) have over five years of working experience in projects, suggesting they possess a solid understanding of project practice and can make effective judgments. Meanwhile, early participants and late respondents are compared in terms of their profession, years of experience, organizational affiliation, and types of projects. The nonresponse bias in this study was not evident, as there were no significant differences between the two groups.

**Table 1.** Demographic characteristics of the sample.

| Features | Category | Quantity | Percentage |
|---|---|---|---|
| Project Type | General construction work | 119 | 54.3 |
| | Oil and gas engineering | 2 | 0.9 |
| | Transportation Engineering | 27 | 12.3 |
| | Power engineering | 22 | 10.0 |
| | Hydraulic engineering | 20 | 9.1 |
| | Industrial plants | 9 | 4.1 |
| | Other | 20 | 9.1 |
| Workplace | Owners | 38 | 17.4 |
| | Contractors | 113 | 51.6 |
| | Consultancy | 51 | 23.3 |
| | Other | 17 | 7.8 |
| Role in the project | Senior Management | 28 | 10.6 |
| | Middle managers | 64 | 29.2 |
| | Executive level | 109 | 49.8 |
| | Other | 18 | 8.2 |
| Education | Doctorate | 4 | 1.8 |
| | Graduates | 67 | 30.6 |
| | Undergraduates | 136 | 62.1 |
| | High School and below | 12 | 5.5 |
| Work experience | <5 years | 91 | 41.6 |
| | 6–10 years | 71 | 32.4 |
| | 11–15 years | 41 | 18.7 |
| | >15 years | 16 | 7.3 |

*3.3. Common Method Bias*

There is a serious concern about common-method bias since the data were obtained through self-reporting questionnaires. To mitigate this, the items were adjusted to ensure participants felt confident that their responses were confidential and anonymous. A unique identifier code was assigned to each participant for data collection and analysis, preventing personal identification. Additionally, several items were reverse-scored to avoid directly addressing negative aspects. In Harman's single-factor test, no single factor can explain more than 50% of the variance, indicating there was no common method bias. In addition, acknowledging the potential limitations of traditional tests, Chi proposed the application of the variance inflation factor (VIF) as an additional corrective measure [35]. In the multicollinearity test conducted during regression, VIF was less than 3, and no obvious collinearity problem was found among the predictors, which ensured the reliability of the analysis results.

*3.4. Construct Reliability and Validity Measures*

SPSS 24 and AMOS 24 software were applied to analyze the data. Assess the measurement's internal consistency reliability by calculating Cronbach's Alpha for each construct. As can be seen in Table 2, the Cronbach's Alphas are between 0.7 and 0.9, and the CITC values of the measurement items for the six latent variables are all greater than 0.5, indicating

that the scale has a high level of reliability, and there is a high degree of internal consistency in the scale for the latent variables. A confirmatory factor analysis was performed on these critical variables through AMOS 24, and the analysis results are shown in Tables 3 and 4: $\chi^2/df$ = 1.340, $p < 0.01$, which reaches the level of significance. As such, the measurement model and empirical data show significant differences in their covariance matrices. Due to the limitations of the chi-square test, other indicators need to be validated. The GFI and AGFI were 0.927 and 0.896, respectively, both exceeding 0.8. The RMSEA was 0.039, <0.05. As a result, the model was acceptable. The IFI, CFI, and NFI are 0.984, 0.984, and 0.939, respectively, all greater than 0.90. Thus, this study's factor model fits well and has high validity. For each latent variable, the standardized factor-loading coefficients were greater than 0.7, indicating that the model has convergent validity. Good convergent validity is indicated when the average variance extracted (AVE) value is greater than 0.5, while the square root of the AVE is greater than the non-diagonal correlation coefficient, demonstrating superior discriminant validity, as shown in Table 4. Consequently, these 18 observations can be used as a measure of inter-organizational relationship cognition, organizational behaviors, and value added in projects.

**Table 2.** Measures of reliability and validity assessment.

| | Item | CITC | Cronbach's Alpha If Item Deleted | Cronbach's Alpha | AVE | CR |
|---|---|---|---|---|---|---|
| Trust | T1 | 0742 | 0.849 | | | |
| | T2 | 0.786 | 0.811 | 0.879 | 0.710 | 0.880 |
| | T3 | 0.771 | 0.825 | | | |
| Shared vision | SV1 | 0.727 | 0.802 | | | |
| | SV2 | 0.782 | 0.782 | 0.856 | 0.667 | 0.857 |
| | SV3 | 0.714 | 0.812 | | | |
| In-role behavior | IB1 | 0.784 | 0.839 | | | |
| | IB2 | 0.792 | 0.832 | 0.888 | 0.727 | 0.889 |
| | IB3 | 0.769 | 0.852 | | | |
| Extra-role behavior | EB1 | 0.751 | 0.782 | | | |
| | EB2 | 0.730 | 0.800 | 0.857 | 0.668 | 0.858 |
| | EB3 | 0.711 | 0.818 | | | |
| Opportunistic behavior | OB1 | 0.568 | 0.670 | | | |
| | OB2 | 0.649 | 0.574 | 0.790 | 0.519 | 0.757 |
| | OB3 | 0.509 | 0.737 | | | |
| Value added | VA1 | 0.700 | 0.745 | | | |
| | VA2 | 0.714 | 0.728 | 0.826 | 0.619 | 0.829 |
| | VA3 | 0.637 | 0.806 | | | |

T: trust; SV: shared vision; IB: in-role behavior; EB: extra-role behavior; OB: opportunistic behavior; VA: value added; CR: composite reliability; AVE: average variance extracted.

**Table 3.** Evaluation results of discriminant validity.

| Variables | T | SV | IB | EB | OB | VA |
|---|---|---|---|---|---|---|
| T | 0.710 | | | | | |
| SV | 0.710 *** | 0.667 | | | | |
| IB | 0.786 *** | 0.673 *** | 0.727 | | | |
| EB | 0.571 *** | 0.495 *** | 0.507 *** | 0.668 | | |
| OB | −0.333 *** | −0.297 *** | −0.329 *** | −0.259 *** | 0.519 | |
| VA | 0.589 *** | 0.542 *** | 0.548 *** | 0.434 *** | −0.290 *** | 0.619 |
| Square root of AVE | 0.843 | 0.817 | 0.853 | 0.817 | 0.720 | 0.787 |

Note: T: trust; SV: shared vision; IB: in-role behavior; EB: extra-role behavior; OB: opportunistic behavior; VA: value added; boldface signifies that the square roots of AVE are greater than the off-diagonal correlations. *** $p < 0.001$.

**Table 4.** Evaluation results of fit indices.

| Fit Indices | Indicators | Fit Standard | Values | Fit Condition |
|---|---|---|---|---|
| Absolute fit indices | $\chi^2/df$ | <2 | 1.340 | √ |
| | GFI | >0.8 | 0.927 | √ |
| | AGFI | >0.8 | 0.896 | √ |
| | RMSEA | <0.05 | 0.039 | √ |
| | RMR | <0.05 | 0.040 | √ |
| Contracted fit indices | PNFI | >0.5 | 0.736 | √ |
| | PCFI | >0.5 | 0.771 | √ |
| Relative fit indices | NFI | >0.9 | 0.939 | √ |
| | CFI | >0.9 | 0.984 | √ |
| | IFI | >0.9 | 0.984 | √ |
| | TLI | >0.9 | 0.979 | √ |

Note: $\chi^2/df$: chi-square/degree of freedom; GFI: goodness-of-fit index; AGFI: adjusted goodness-of-fit index; NFI: normal fit index; IFI: incremental fit index; CFI: comparative fit index; RMSEA: root mean square error of approximation; RMR: root mean square residual; PNFI: parsimonious normed fit index; PCFI: parsimonious comparative fit index; TLI: Tucker-Lewis Index; √: fit condition is good.

## 4. Model Development and Analysis

### 4.1. Establishment of the SEM Model and Model Validation

To understand the factors that influence inter-organizational relationship cognition in the value added, a structural equation model was constructed to examine the static causal relationships between different variables [27]. Due to the statistical tests that the SEM provides, it is capable of guaranteeing data consistency. Additionally, the developed model will provide a platform for further modeling of the FCM [24]. Models of SEM can be classified into two types: measurement models and structural models. Structural models reveal the relationships between potential variables, while measurement models address the reliability and validity of potential variable measurements [25]. Based on the literature review, after identifying the variables and measurement items of inter-organizational relationship cognition, organizational behaviors, and value-added, the SEM model was constructed using the AMOS 24 with six variables, including two independent variables, three mediating variables, and one dependent variable. Predictions made by the model were evaluated for acceptability and accuracy [66]. Each variable was measured by three observed variables with high reliability and validity. The path coefficients between the variables in the SEM model are shown in Tables 5 and 6. The model is valid if the *p*-value is less than 0.05. The path coefficients of trust on in-role behavior and extra-role behavior were 0.492 and 0.332, indicating a positive effect of trust on in-role behavior and extra-role behavior. In contrast, the *p*-value of trust on opportunistic behavior was 0.072 > 0.05, which suggests that trust has no significant effect on opportunistic behavior. Increased trust among participants raises value expectations, fostering a belief in the possibility of regulated interactions and motivating in-role behavior. Trust also enhances mutual appreciation and cohesion, encouraging cooperative actions to meet partner expectations. However, trust alone may not be enough to prevent opportunistic behavior, especially in the complex, uncertain environments of construction projects, where participants tend to prioritize their own interests. Building and maintaining trust is challenging, as its impact on extra-role and opportunistic behaviors can be diminished by information asymmetry and conflicting interests. It was found that shared vision had 0.445, 0.341, and −0.449 path coefficients for in-role behavior, extra-role behavior, and opportunistic behavior, respectively, indicating that it was positively correlated with both in-role and extra-role behavior, but negatively correlated with opportunistic behavior. In engineering projects, where disputes are common, a shared vision acts as a binding mechanism that improves coordination and fosters understanding, forming a strong foundation for collaboration. This connection between shared vision and cooperative behavior is evident as common goals and decision-making processes lead to mutual respect and active information exchange.

As a strategic framework, a shared vision can effectively reduce opportunistic behavior during project execution. Hence, inter-organizational relationship cognition can encourage cooperative behavior among organizations and discourage opportunistic behavior, thereby increasing value added. Table 6 presents the results of the GOF analysis of the SEM model, which indicates that $\chi^2/df$ is 1.425 (less than 2) with a *p*-value of 0.000, indicating that the model has reached significance, and that the covariance matrix of the measurement model differs significantly from the empirical data. In order to test the model's fit, it is necessary to refer to other indicators. For the absolute fit indices, the RMR value was 0.046 (<0.05); the value of RMSEA was 0.044 (<0.05); and the values of GFI and AGFI were 0.919 and 0.889 (>0.8), respectively. For the relative fit indices, the NFI was 0.932 (>0.8), and the CFI, IFI, and TLI were 0.979, 0.9795, and 0.974 (>0.9), correspondingly. For the parsimonious fit index, PNFI and PCFI were 0.762 and 0.799, respectively, both greater than 0.5. Clearly, all 11 adaptive indicators meet the critical value requirement. The SEM model fits well. Therefore, the SEM model can be used to construct the FCM model.

**Table 5.** Analysis of path effects among constructs in the SEM model.

| Causal Path | Standard Coefficient | *p*-Value | Interpretation |
|---|---|---|---|
| T→IB | 0.492 | *** | Supported |
| T→EB | 0.332 | * | Supported |
| T→OB | −0.284 | 0.072 | Not supported |
| SV→IB | 0.445 | *** | Supported |
| SV→EB | 0.341 | * | Supported |
| SV→OB | −0.449 | ** | Supported |
| IB→SV | 0.479 | *** | Supported |
| EB→SV | 0.161 | * | Supported |
| OB→SV | −0.341 | *** | Supported |
| T→IB→SV | 0.236 | *** | Supported |
| T→EB→SV | 0.053 | * | Supported |
| T→OB→SV | 0.097 | 0.076 | Not supported |
| SV→IB→SV | 0.213 | *** | Supported |
| SV→EB→SV | 0.055 | * | Supported |
| SV→OB→SV | 0.153 | * | Supported |

Note: T: trust; SV: shared vision; IB: in-role behavior; EB: extra-role behavior; OB: opportunistic behavior; * $p < 0.05$. ** $p < 0.01$. *** $p < 0.001$.

**Table 6.** Results of the goodness-of-fit analysis for SEM model.

| Fit Indices | Indicator | Fit Standards | Value | Test Results |
|---|---|---|---|---|
| Absolute fit indices | $\chi^2/df$ | <2 | 1.425 | √ |
| | GFI | >0.8 | 0.919 | √ |
| | AGFI | >0.8 | 0.889 | √ |
| | RMSEA | <0.05 | 0.044 | √ |
| | RMR | <0.05 | 0.046 | √ |
| Parsimony fit indices | PNFI | >0.5 | 0.762 | √ |
| | PCFI | >0.5 | 0.799 | √ |
| Relative fit indices | NFI | >0.8 | 0.932 | √ |
| | CFI | >0.9 | 0.979 | √ |
| | IFI | >0.9 | 0.979 | √ |
| | TLI | >0.9 | 0.974 | √ |

Note: $\chi^2/df$: chi-square/degree of freedom; GFI: goodness-of-fit index; AGFI: adjusted goodness-of-fit index; NFI: normal fit index; IFI: incremental fit index; CFI: comparative fit index; RMSEA: root mean square error of approximation; RMR: root mean square residual; PNFI: parsimonious normed fit index; PCFI: parsimonious comparative fit index; TLI: Tucker-Lewis Index; √: test result is good

## 4.2. Establishment of FCM Model

The basis of FCM model building lies in identifying the appropriate weights for indicating the relative strength of causal relationships between distinct concepts [67]. It

is ambiguous and subjective to determine a weight based on expert knowledge, whereas SEM is a reliable method for identifying and assessing causal relationships among various factors using questionnaire data in a robust manner [27,68]. Therefore, it enhances the accuracy and reliability of the FCM model by ensuring that the estimated weights are reliable. On the basis of the SEM model, the FCM model is built to examine the evolution of inter-organizational relationship cognition and value added. In the established SEM model, inter-organizational relationship cognition has a significant impact on value added. Validating the SEM model is also necessary in order to eliminate errors or bias during its establishment, which results in a reliable and valid model. Thus, the path coefficients can be used to evaluate causal linkages between inter-organizational relationship cognition and value added. In the FCM model, the path coefficient values are used as inputs [69]. The conceptual nodes of the FCM model are divided into two categories: cause and target concept [24]. Concept weights are derived from influence path and correlation values in Table 5. Thus, the cause nodes consist of trust, shared vision, in-role behavior, extra-role behavior, and opportunistic behavior, and the target node is the value added. The FCM model is constructed and shown in Figure 2.

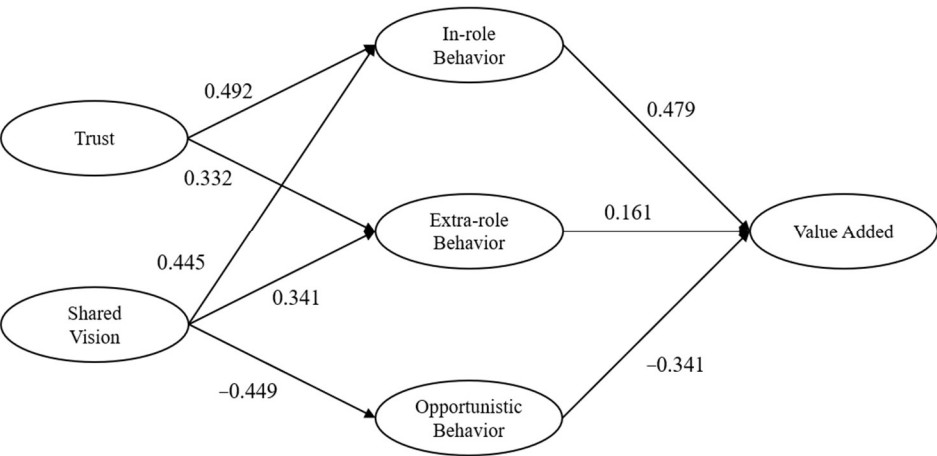

**Figure 2.** FCM model for value added to projects.

The inference mechanism of the FCM is implemented by modeling the dynamic process of causality between its nodes as it evolves over time [70]. The model analyses in this study include predictive, diagnostic, and mixed analyses. Predictive analyses aim to examine the extent to which factors affect value added and quantify the role of each variable in value addition. Diagnostic analysis seeks to identify the root causes that are most likely to affect the value added, primarily through the backward derivation of the FCM model. In the mixed analysis, the main objective is to predict future state changes using the dynamic characteristics of the FCM, as well as diagnose potential system problems and influences by simulating the system's behavior and feedback mechanisms [71].

### 4.2.1. Predictive Analysis

Predictive analysis is the process of speculating and forecasting the future state or behavior of a system using models, which aims to predict the evolution of future conceptual nodes to explore the extent to which influencing factors will impact value added. In the FCM model, predictive analyses enable the quantification of the role of each variable in value added. This process can be described as a direct influence relationship between the cause node and the target node.

In predictive analyses, when the effect of only one conceptual node is examined, it is necessary to assume the state value of this node as I while making the state values of the other conceptual nodes as 0, as well as monitoring the evolution process and stabilizing the value added. In each scenario, the initial state of a specific influencing factor was set to $-1$ (very small), $-0.5$ (smaller), $0.5$ (larger), and $1$ (very large). Using causal reasoning

and iterative calculations, a simulated value of the engineering project value-added effect is determined. The effect of inter-organizational relationship cognition and changes in organizational behavior on value added is shown in Figure 3 and the final convergence values of the steady state of the engineering project value-added effect after different scenario iterations are shown in Table 7.

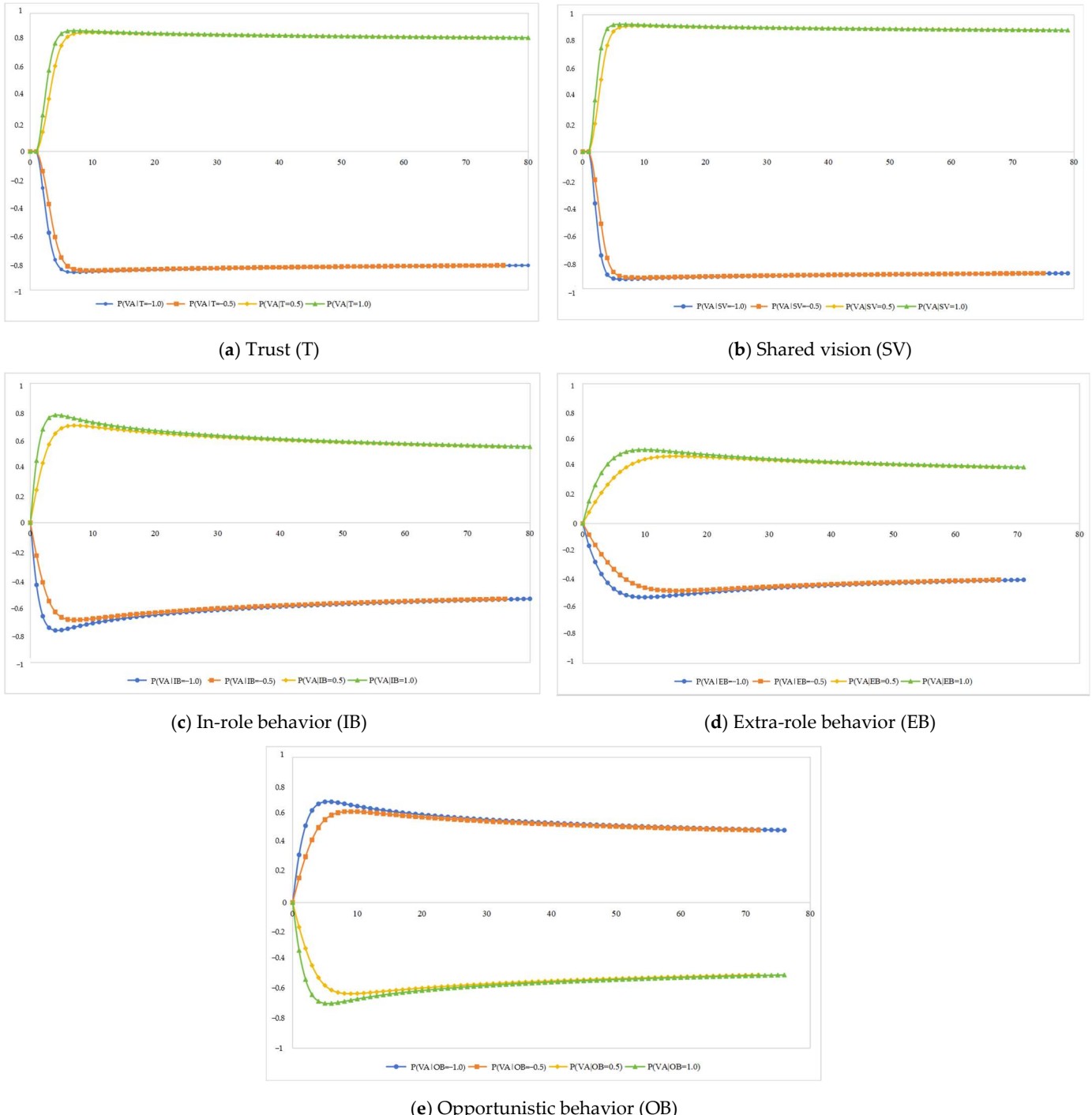

(**a**) Trust (T)

(**b**) Shared vision (SV)

(**c**) In-role behavior (IB)

(**d**) Extra-role behavior (EB)

(**e**) Opportunistic behavior (OB)

**Figure 3.** Impacts of variations in the influencing variables on the value added (VA) in different scenarios, when the variable V is set to be a value of −1, −0.5, 0.5, and 1: (**a**) trust (T); (**b**) shared vision (SV); (**c**) in-role behavior; (**d**) extra-role behavior (EB); (**e**) opportunistic behavior (OB).

**Table 7.** Stable values of project value-added effects after different scenario iterations in the projection analysis.

| Items | P(VA ǀ i = 1.0) | P(VA ǀ i = 0.5) | P(VA ǀ i = −0.5) | P(VA ǀ i = −1.0) |
|:---:|:---:|:---:|:---:|:---:|
| T | 0.8229 | 0.8228 | −0.8228 | −0.8229 |
| SV | 0.8867 | 0.8866 | −0.8866 | −0.8867 |
| IB | 0.5447 | 0.5445 | −0.5445 | −0.5447 |
| EB | 0.4026 | 0.4023 | −0.4023 | −0.4026 |
| OB | −0.4977 | −0.4974 | 0.4974 | 0.4977 |

Note: T: trust; SV: shared vision; IB: in-role behavior; EB: extra-role behavior; OB: opportunistic behavior; VA: value added.

Taking one scenario as an example, it simulates the impact of changes in trust on value added. Figure 3a and Table 7 show that, for T = −1, after 80 iterations, the value added reaches a steady state, with a convergence value of −0.8229; for T = −0.5, after 76 iterations, the value added reaches a steady state, with a convergence value of −0.8228; for T = 0.5, after 76 iterations, the value added reaches a steady state, with a convergence value of 0.8228; and for T = 1, after 80 iterations, the value added reaches a steady state, with a convergence value of 0.8229. This shows that trust has facilitating effects on the value added. When trust was initially set to 1 (or −1), a higher (or lower) level of value added was achieved than when trust was initially set to 0.5 (or −0.5). However, the magnitude of the increase (or decrease) was insignificant. The results suggest that high levels of value added cannot be developed without intervening in trust. Moreover, the more substantial interventions on trust are not better; the most effective and economical intervention is to set the trust at 0.5. It can be seen that trust and value added have a positive correlation.

As shown in Figure 3b, shared vision has significant facilitating effects on the value added. The initial state value of 1 (or −1) for shared vision resulted in a higher (or lower) level of value added compared to a value of 0.5 (or −0.5). However, the magnitude of the increase (or decrease) was insignificant. If shared vision and value added were initially 1 (or −1), there would be a higher (or lower) level of value added than if they were initially 0.5 (or −0.5), which would indicate a substantial change in the value added. These results show that when the intervention of shared vision is insufficient, the value-added effect is not much different from the effect of no intervention, indicating that the intervention effect is poor, which means that the most effective and cost-effective intervention would be to raise the level of shared vision to 0.5.

According to Figure 3c,d, in-role behavior and extra-role behavior have a positive effect on the engineering project value added. With the gradual increase in the in-role behavior or extra-role behavior state value, the value-added effect is significantly enhanced. When IB = 0.5 (or −0.5), there is little change in value addition results compared to IB = 1 (or −1). However, when IB = 0.5 (or −0.5), with an increase in the number of iterations, the value-added effect reaches the highest (lowest) point after about 10 iterations, then gradually decreases (increases) in a small way, and finally stabilizes. This trend is more obvious when IB = 1 (or −1). This means that intervening in-role behavior or extra-role behaviors can significantly increase value added in the short term, but the effect on the engineering project value added gradually stabilizes over time. Therefore, the most effective and economical approach intervention is set in-role and extra-role behavior to 0.5, but this intervention is a short-term strategy.

As shown in Figure 3e, opportunistic behavior has a negative effect on the engineering project value added. With the increasing state value of opportunistic behavior, the effect of value added is significantly weakened. Compared with OB = 1 (or −1), when OB = 0.5 (or −0.5), the result of the value added has little change. However, when OB = 0.5 (or −0.5), with an increase in the number of iterations, the value-added effect reaches the lowest (highest) point after about 15 iterations, and then gradually decreases (rises) in a small way, and finally stabilizes. This trend is more obvious when OB = 1 (or −1). This means that by intervening in opportunistic behavior, the value added can be significantly increased in the

short term, but the growth effect will gradually diminish over time. Therefore, the most effective and economical way is to set opportunistic behavior to −0.5, but this intervention is a short-term strategy.

Based on the above analysis, moderate interventions in project participants' cognitions, as well as behaviors (i.e., T = 0.5, SV = 0.5, IB = 0.5, EB = 0.5, and OB = −0.5), can effectively contribute to the engineering project value addition. For further comparison, T = 0.5, SV = 0.5, IB = 0.5, EB = 0.5, OB = −0.5, and simulation analysis are conducted. Figure 4 illustrates the results. Clearly, moderate interventions on shared vision and trust enhance value-added effects significantly. When the FCM model moves to a steady state, P(VA | EB = 0.5) < P(VA | OB = −0.5) < P(VA | IB = 0.5) < P(VA | T = 0.5) < P(VA | SV = 0.5). Therefore, among the five influencing factors in this study, trust and shared vision are the most critical influencing factors, and addressing these two aspects through appropriate interventions can effectively add value to the project. In-role behaviors, opportunistic behaviors, and extra-role behaviors are the next most important influencing factors, and moderate interventions can contribute to value addition to some extent.

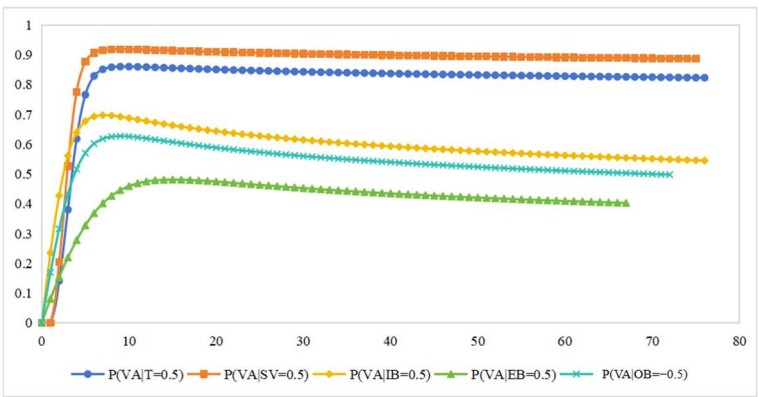

**Figure 4.** Evolutionary curve of value-added effects under appropriate cognition and behavioral interventions.

### 4.2.2. Diagnostic Analysis

Diagnostic analysis aims to identify the root causes that are most likely to have an impact on value added of the engineering project, mainly through the reverse derivation of the FCM model. Figure 5 shows a specific model.

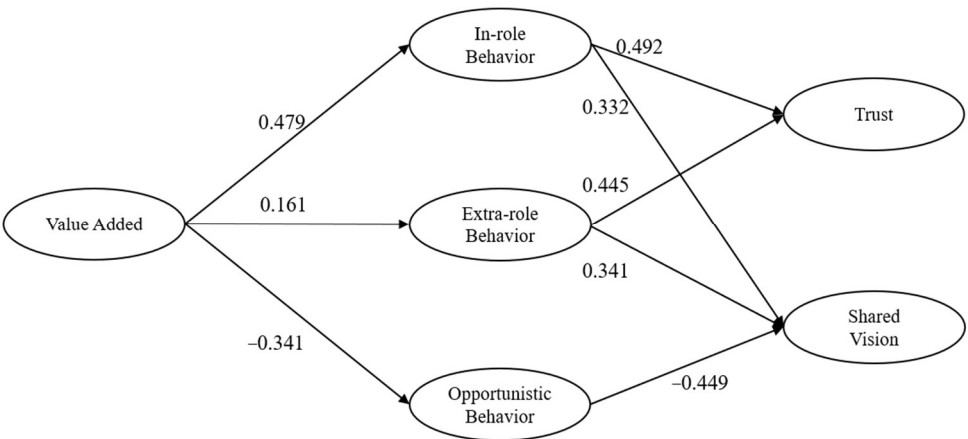

**Figure 5.** FCM model for diagnostic analysis.

To examine the impact of inter-organizational relationship cognition on engineering project value-added effects, all factors' initial state values were set to 0. The state values of value added were set as −1 (very small), −0.5 (small), 0.5 (large), and 1 (very large),

respectively. Then, we simulated evolutionary trends in inter-organizational relationship cognition and organizational behavior. The state value of the value-added effect is updated each iteration until it reaches a steady state after a certain number of computations. Figure 6 shows how the value-added changes affect each factor in the diagnostic analysis, and Table 8 shows the convergence values for each factor after multiple iterations in different scenarios. Each curve represents the degree to which every factor is sensitive to changes in the value added, and the point at which each curve reaches a steady state represents how much each factor is affected by the changes in the value added. As can be seen from Figure 6, when the effect of engineering project value added is in a very poor or poor state (i.e., VA = −1 or VA = −0.5), opportunistic behavior initially increases and then stabilizes, while other factors initially decrease and then stabilize. When the value-added effect is in a good or very good state (i.e., VA = 0.5 or VA = 1), trust, shared vision, in-role behavior, and extra-role behavior initially increase and then stabilize, while opportunistic behavior initially decrease and then stabilize. Although the value-added effect may be at different stages, it is clear that the changes in trust and shared vision between organizations in engineering projects are the most significant, indicating that they are likely fundamental factors influencing the value added. Among the various factors, trust and shared vision converge the fastest, which means they are more sensitive to changes in the value added. In other words, when promoting the engineering project value added, it is essential to first assess the cognition of trust and shared vision within the organization and then intervene in these cognitions accordingly to achieve value-added goals more effectively.

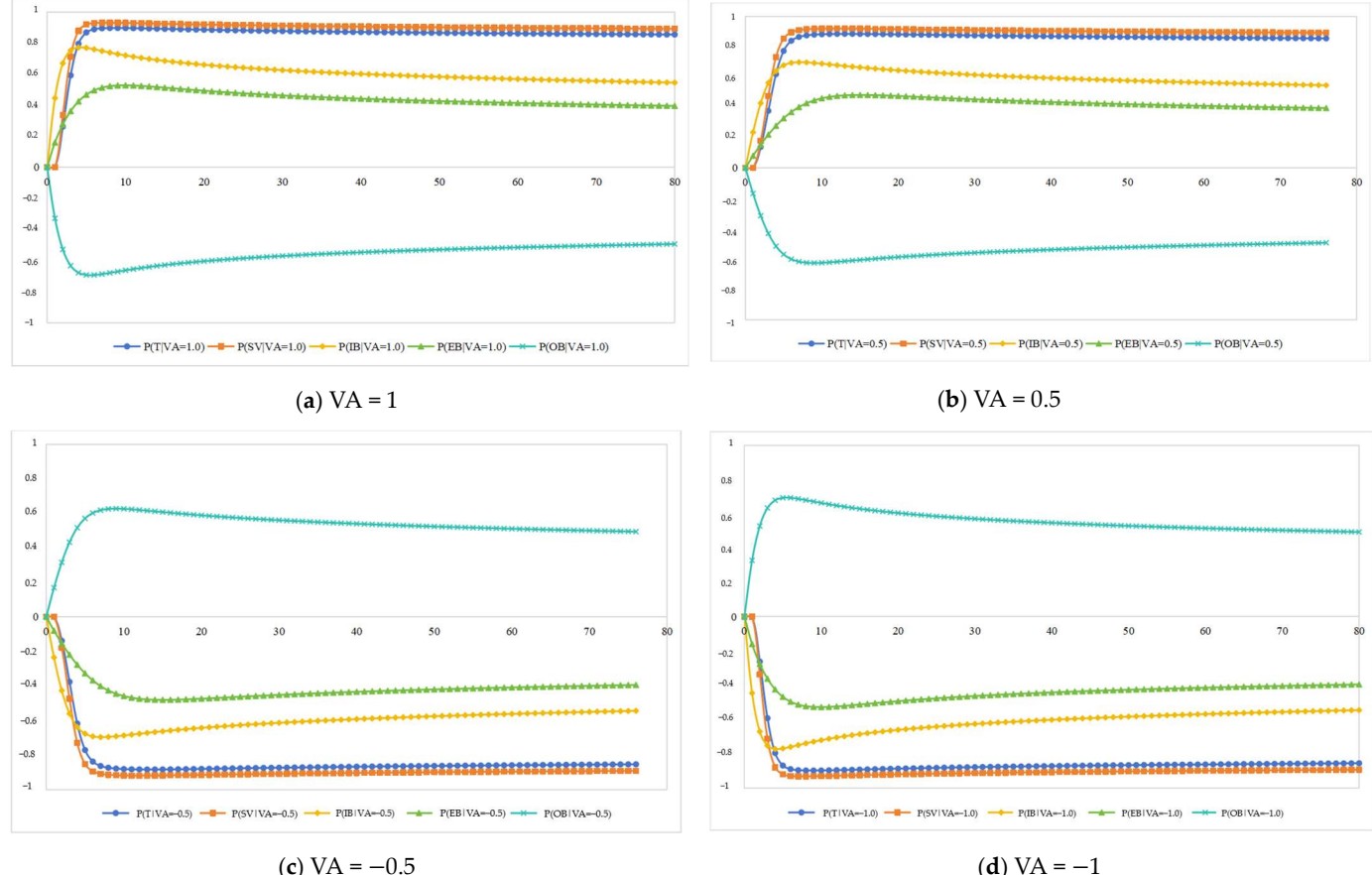

(**a**) VA = 1

(**b**) VA = 0.5

(**c**) VA = −0.5

(**d**) VA = −1

**Figure 6.** Impacts of variations in value added on the influencing variables V (V = T, SV, IB, EB or OB) when value added (VA) is set to be: (**a**) VA = 1; (**b**) VA = 0.5; (**c**) VA = −0.5; (**d**) VA = −1.

**Table 8.** Convergence values for each variable after multiple iterations in different scenarios in diagnostic analysis.

| Target Node | P(i | VA = 1.0) | P(i | VA = 0.5) | P(i | VA = −0.5) | P(i | VA = −1.0) |
|---|---|---|---|---|
| T | 0.8543 | 0.8542 | −0.8542 | −0.8543 |
| SV | 0.8917 | 0.8917 | −0.8917 | −0.8917 |
| IB | 0.5447 | 0.5445 | −0.5445 | −0.5447 |
| EB | 0.3949 | 0.3947 | −0.3947 | −0.3949 |
| OB | −0.4939 | −0.4936 | 0.4936 | 0.4939 |

Note: T: trust; SV: shared vision; IB: in-role behavior; EB: extra-role behavior; OB: opportunistic behavior; VA: value added.

### 4.2.3. Hybrid Analysis

By setting the initial values of each influencing factor simultaneously, hybrid analysis continuously iterates value added and the influencing factors. As well, the trend in value added can be compared by adjusting state values of different factors, which provides a theoretical basis for designing cost-effective interventions.

To begin with, the most unfavorable engineering project value added scenario is simulated in which all the influencing factors and value added are at unfavorable levels, i.e., T = −1, SV = −1, IB = −1, EB = −1, OB = 1, and VA = −1. The FCM model reaches a steady state where value added convergence value to −0.9035, which indicates that value added is extremely ineffective. According to the predictive analyses, trust and shared vision are the most critical factors. In order to promote the value added, both trust and shared vision should be controlled at the 0.5 level. Based on the diagnostic analyses, trust and shared vision may be the root cause of adding value to the project. Therefore, the hybrid analysis focused on interventions related to trust and shared vision.

However, a single intervention scenario is designed to adjust the state value of trust and shared vision to 0.5, respectively, whereby the value-added effect is −0.6830 and 0.3191, respectively. It appears that a single intervention on trust can enhance the value-added effect of the project, but the enhancement is relatively small, whereas a single intervention on shared vision has a significant impact. As a result, promoting value added is not feasible by intervening only in one aspect in practice.

Finally, when considering the integrated intervention scenarios, Table 9 summarizes the adjustments to the initial state values of trust, shared vision, in-role behavior, extra-role behavior, and opportunistic behavior in the various integrated intervention scenarios, as well as the convergence values of value added in the corresponding scenarios. Table 9 indicates that simultaneous interventions on trust, shared vision, in-role behavior, and opportunistic behavior (integrated intervention 6) were most effective when the convergence value of the value added steady state was 0.8596, implying a high level of value added. Integrated intervention 1 worked well, with the value-added convergence value at 0.9035 when the model stabilized; however, in the other integrated interventions, the convergence values of value added ranged between −0.3 and −0.6, suggesting an ordinary degree of VA, and the integrated intervention strategy was not effective.

**Table 9.** Setting of state values for different integrated intervention scenarios and convergence values of project value added.

| Integrated Intervention Scenarios | T | SV | IB | OB | VA Convergence Value |
|---|---|---|---|---|---|
| 1 | 0.5 | 0.5 | −1 | 1 | 0.9035 |
| 2 | 0.5 | −1 | 0.5 | 1 | −0.6374 |
| 3 | 0.5 | −1 | −1 | −0.5 | −0.6830 |
| 4 | −1 | 0.5 | 0.5 | 1 | 0.3196 |
| 5 | −1 | 0.5 | −1 | −0.5 | 0.3191 |
| 6 | 0.5 | 0.5 | 0.5 | −0.5 | 0.9280 |

Note: T: trust; SV: shared vision; IB: in-role behavior; OB: opportunistic behavior; VA: value added.

The results of the analysis of the single intervention scenario show that in order to add value to an engineering project, it is necessary to intervene in the inter-organizational relationship cognition of the project participants, in addition to focusing on interventions on in-role and opportunistic behaviors. In Table 9, the value-added convergence value is greatest, i.e., most effective, when inter-organizational relationship cognition and organizational behavior are simultaneously addressed. It is, however, necessary to consider management costs and select cost-effective strategies for actual projects. According to integrated intervention 1 and 6, there is little difference in the convergence values of the value added. Considering the cost–benefit principle, it appears that a strategy of combined interventions on trust and shared vision could be selected in practice.

## 5. Discussion

The SEM results confirm the significant impact of inter-organizational relationship cognition on engineering project value added. However, in the engineering industry, value added is an ongoing process essential for the sustainability of an organization and its long-term future viability. As a result, cross-sectional data cannot accurately reflect how value added has evolved over time in terms of how interorganizational relationships are perceived. By offering theoretical and practical insights into the dynamic progression and overall improvement of value added throughout a project lifecycle, the simulation analysis enhances existing research on the formation, development, and intervention of value added. This study investigates the potential dynamic mechanisms through which inter-organizational relationship cognition influences engineering project value added combining SEM and FCM. This approach enables the formulation of appropriate management strategies to optimize project outcomes.

### 5.1. Static Relationships between Inter-Organizational Relationship Cognition and Value Added

First, trust positively influences both in-role and extra-role behaviors. Contrary to expectations, trust does not have a significant impact on opportunistic behavior. The findings regarding the relationship between trust and in-role behavior align with the results of Yang's research, which emphasizes the critical role of trust in the behavioral outcomes of in-role behavior [72]. As mutual trust increases participants' value expectations, they generally perceive that normative interactions are possible, thereby motivating in-role behavior. Additionally, trust is a psychological factor that guides partners to appreciate each other, enhancing their cohesion. A trusted party is more likely to engage in cooperative behavior to reciprocate their partner's expectations. However, the value expectations and cohesion generated by trust are not sufficient to curb opportunistic behavior. One possible reason is that construction projects often involve complex and dynamic environments, where participants face numerous uncertainties. In such situations, their natural inclination is to prioritize their own interests and priorities. Therefore, building and maintaining trust is a challenge. Even when there is a certain degree of trust between participants, potential information asymmetry and conflicts of interest can offset its impact on extra-role behavior and opportunistic behavior.

Secondly, shared vision positively influences cooperative behavior, consistent with the findings of Koh [73]. Given the frequent disputes in construction projects, a shared vision can be seen as a bonding mechanism that enhances coordination efficiency. It also fosters understanding and lays a solid foundation for cooperation. This indicates a beneficial link between a shared vision and cooperative behavior. Similarly, the mitigation of opportunistic behavior by a shared vision is also validated. This result aligns with Wong's findings on the relationship between a shared vision and opportunistic behavior [74]. Due to common goals and a mutual understanding of decisions, collaborators respect each other and actively engage in comprehensive information exchange. Shared vision, as a top-level framework, can effectively curb opportunistic behavior among partners during project implementation. The findings also extend Chi's research by revealing the potential

psychological and participatory behavior relationships between a shared vision and the value added [35].

Thirdly, cooperative behaviors (both in-role and extra-role behaviors) positively impact the value added, while opportunistic behaviors negatively impact the value added. These conclusions align with existing research findings in other contexts. Based on the empirical results, it can be concluded that participants' in-role and extra-role behaviors are key drivers of engineering project value added. To achieve value added, attention must be paid to each partner's behavior. In-role and extra-role behaviors ensure the achievement or even overachievement of project goals, while opportunistic behaviors undoubtedly cause losses. Project managers should improve traditional methods that rely on suppressing partners' opportunistic behaviors to achieve project goals. Striking a balance between cooperative behavior and opportunistic behavior becomes crucial. This balance ensures that the positive effects of cooperative behavior remain unaffected while preventing opportunistic behavior from eroding collective interests.

Finally, inter-organizational relationship cognition can encourage cooperative behaviors among participants and inhibit their opportunistic behaviors, thereby promoting the value added. Cooperative behavior serves as the link between trust and value added, realizing the positive impact of trust on value added. In contrast, the mediating role of opportunistic behavior is not significant. The relationship between a shared vision and value added is mediated by both cooperative behavior and opportunistic behavior. It is noteworthy that both trust and a shared vision positively impact the value added, but a shared vision plays a greater role in influencing participants' behaviors. Therefore, the role of a shared vision should be emphasized. These research conclusions provide new insights into value added in project management.

### 5.2. Dynamic Relationships between Inter-Organizational Relationship Cognition and Value Added

It is consistent with empirical research that interorganizational relationship cognition, organizational behavior, and value added have a significant causal relationship. The study indicates that the impact of various inter-organizational relationship cognition and organizational behavior variables on value added differs. It has also been shown that applying multiple intervention strategies at the same time is more effective than using one intervention strategy alone. In order to enhance the value added in engineering projects, multiple relevant factors must be prioritized and reasonable strategies developed.

In engineering projects, inter-organization relationship cognition and organizational behavior affect the value added to the project in varying degrees. Based on research, shared vision, trust, in-role behavior, opportunistic behavior, and extra-role behavior are the factors influencing engineering project value added. A balance must be struck between the intensity of interventions and their costs and effects in order to determine the optimal strength of interventions for promoting value added. An analysis of the state values of trust, shared vision, in-role behavior, extra-role behavior, and opportunistic behavior has shown that these values can be maintained at 0.5, 0.5, 0.5, 0.5, and $-0.5$, respectively, resulting in a higher value added and a favorable cost–benefit ratio. Diagnostic analysis has revealed the fundamental causes of promoting the engineering project value added. Ranked by likelihood, these causes are shared vision, trust, in-role behavior, opportunistic behavior, and extra-role behavior. Therefore, in engineering projects, the most likely fundamental cause of project value added is the maintenance of good trust and a shared vision among project participants. Interventions in the cognition of inter-organizational relationships should be emphasized. In-role behavior, extra-role behavior, and opportunistic behavior are also important factors in promoting project value added. Achieving consensus on trust and shared vision among project participants provides an appropriate cognition that offers the opportunity to realize value added.

It is crucial that a comprehensive intervention focuses on trust and shared vision, taking into consideration a wide range of organizational behaviors among project participants in order to raise the value added in engineering projects. Enhancing inter-organizational

trust and shared goals has been proven to be a key method for achieving value added. In practice, efforts should be made to strengthen communication and establish good inter-organizational relationship cognition. Additionally, comprehensive intervention measures are more effective than single interventions in enhancing the value added. Considering cost-effectiveness, it is recommended to first focus on building inter-organizational trust and shared vision to maximize the return on investment. When resources are ample, more comprehensive interventions, including adjustments in the cognition of inter-organizational relationship cognition and behaviors, should be considered to further enhance the value added and effectively achieve project goals.

### 5.3. Theoretical Implications

Firstly, this study delved into the intricate interplay between inter-organizational relationship cognition and value added while also considering the mediating role of participants' organization behavior. Unlike previous studies that examined cooperative and opportunistic behaviors in isolation, this study takes a novel approach by considering both behaviors together and then parsing cooperative behavior into in-role and extra-role manifestations. This detailed analysis allows project participants to find empirical support in construction practice, validating the practical impact of inter-organizational relationship cognition on achieving value added.

Secondly, the study reveals that various factors significantly impact the development of value added in engineering projects over time. Understanding the varying degrees of influence of these factors is crucial for fostering value added. Diagnostic analysis shows that a shared vision is likely a fundamental reason why project organizations achieve high levels of value added. This finding is consistent with the previous research, which emphasizes the crucial role of team members in obtaining inter-organizational relationship cognition within project organizations.

Finally, compared with previous studies that only conducted static analyses through SEM and the shortcomings arising from expert subjectivity in the FCM approach, this study combines SEM and FCM to explore the potential influence mechanisms of inter-organizational relationship cognition on the value added in a project, which contributes to the proposal of the corresponding management countermeasures.

### 5.4. Practical Implications

To begin with, to establish trust, project participants should continuously improve their professional and technical capabilities, fostering a mutually dependent cooperative atmosphere during project collaboration. Project managers should focus on fostering trust between internal teams and external partners. Regular communication, transparent workflows, and shared goal-setting can ensure that all participants feel mutually dependent and supported, thereby creating a collaborative atmosphere. When selecting external partners, project managers should prioritize organizations or companies with a strong reputation in the market. This can be achieved by investigating the partner's past projects, customer feedback, and industry certifications. These well-regarded partners often have experienced engineers and architects who provide professional assurance for the project and maintain a responsible attitude throughout the collaboration. Their familiarity with similar projects helps predict potential challenges, optimize project timelines, and ensure smooth project progression. Once strong trust is established, forming lasting partnerships becomes a natural outcome, leading to productive collaboration cycles.

Moreover, to cultivate a shared vision, it is advisable to explicitly include common interests and collaborative goals. Project managers should clearly define the shared objectives, benefit distribution mechanisms, and potential risk management strategies in the contract. This not only helps reduce uncertainties within the project but also ensures that all parties maintain aligned goals and expectations throughout the project's progress, preventing conflicts that could arise from differences in interests. This strategy can significantly encourage teamwork, seamless information exchange, and the emergence of innovative ideas.

Cultivating a shared vision also requires establishing consistent and robust communication channels among participants and project members. This ongoing communication facilitates consensus-building and reinforces the spirit of collaboration. Project managers should regularly organize on-site meetings involving all parties, including experts from various fields, to address technical issues and coordinate resources. These meetings should facilitate timely communication on project progress, existing challenges, and the next steps. This face-to-face interaction is more effective in resolving issues and ensures that all participants' opinions are fully considered. It helps align behaviors, coordinate interests, optimize benefits, and prevent potential setbacks.

### 5.5. Limitations and Future Directions

Despite the significant progress made in this study in studying inter-organizational relationship cognition and value addition, there are still areas of improvement for future research. Firstly, the study sample was primarily composed of construction professionals in China, so it would be interesting and meaningful to expand the sample population by including data in project teams from different parts of the world to see if these same effects are prevalent across cultural boundaries in the future. Additionally, this study views trust as a composite structure, and further research could measure trust on a multidimensional basis. Future studies may expand the sample; refine the cognitive, affective, and behavioral dimensions of trust; and focus on the process of implementing the intervention and evaluating the effects, to provide more practical recommendations. Finally, the study suggests that integrated interventions could increase project value; however, it is necessary to verify the effectiveness of these interventions when applied to actual engineering projects.

## 6. Conclusions

The dynamic linkages between inter-organizational relationship cognition and value added in engineering projects are often overlooked despite the importance of inter-organizational relationship cognition on value added. In consideration of the dynamic interactions among cognition factors, a hybrid method combining SEM and FCM is applied to discern cognition-related factors and model their effect on value added. Using the proposed approach, predictive, diagnostic, and hybrid analyses can be conducted in a variety of contexts, according to the findings. Model analytics clearly demonstrate that the hybrid approach has flexible simulation capabilities to explore the inter-relationships between inter-organizational relationship cognition and value added. Additionally, hybrid approaches can be easily and conveniently applied to practical situations. Surveys of engineering project practitioners are used to determine the SEM model. In this way, the model can be applied to a wide range of practices.

Based on the empirical studies, the following findings have been obtained: (1) Validation of the SEM model reaches a high level, and the path coefficients can be used to build a dynamic model using the FCM model; (2) trust positively influences value added by promoting cooperative behavior, and shared vision positively impacts value added by promoting cooperative behavior and reducing opportunism. (3) The results of the predictive and diagnostic analyses reveal that several factors contributed to engineering project value added, in order of influence: shared vision, trust, in-role behavior, opportunistic behavior, and extra-role behavior. The creation of high levels of value added in project management is most likely influenced by shared vision, as it is the most powerful contributor to value added. (4) The results of the hybrid analyses argue that focusing on building trust and shared vision across organizations ensures maximum input–output ratios.

It is shown that the SEM–FCM approach can be useful in evaluating inter-organizational relationship cognition and value added in engineering projects, and recommendations are offered to increase the probability of success. The managerial implications of our study highlight the importance of identifying the key drivers that influence value added in engineering projects, especially in a highly competitive environment. Recognizing these factors is essential for implementing effective management strategies. First, set clear short-

term and long-term goals to ensure that project participants have a clear understanding of the project's direction, and facilitate information sharing and problem-solving through regular cross-departmental meetings. At the same time, link the project vision to individual performance assessments to motivate the team to embody the core values of the vision in their daily work. Additionally, establish a transparent communication platform, using collaboration tools to update task progress in real time, ensuring that information remains open and transparent. By implementing small-scale trust pilots and trust-building activities, gradually enhance cooperation and trust among project participants. Finally, strengthen the sense of responsibility, ensuring that each participant is accountable for their work and strictly fulfills all commitments made within the project. These strategies will help the team maintain a strong focus on project goals, improve collaboration efficiency, and achieve outstanding results and greater value added in engineering projects.

**Author Contributions:** Conceptualization, M.X. and X.L.; methodology, M.X.; software, Z.B.; validation, M.X. and X.L.; formal analysis, Z.B.; investigation, Y.W.; resources, Y.W.; data curation, M.X.; writing—original draft preparation, M.X.; writing—review and editing, M.X.; visualization, M.X.; supervision, X.L.; project administration, X.L.; funding acquisition, M.X. and X.L. All authors have read and agreed to the published version of the manuscript.

**Funding:** The authors would like to appreciate the reviewers for all helpful comments, and to thank the foundation of Philosophy and Social Science Research in Colleges and Universities in Jiangsu Province (No. 2020SJA1394), Suzhou Science and Technology Plan (Basic Research) Project (SJC2023002), and Postgraduate Research and Practice Innovation Program of Jiangsu Province (KYCX24_3406).

**Data Availability Statement:** The data presented in this study are available on request from the corresponding author.

**Conflicts of Interest:** The authors declare no conflicts of interest.

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
