# Peer review of "Interaction Mechanism between Inter-Organizational Relationship Cognition and Engineering Project Value Added from the Perspective of Dynamic Impact"

_systems, doi:10.3390/systems12090362_

Round 1

Reviewer 1 Report

Comments and Suggestions for Authors

Interaction Mechanism among Inter-Organizational Relationship Cognition and Engineering Project Value Added from the Perspective of Dynamic Impact

  • Abstract:

Main remark is, that there are no research questions or hypotheses presented in the text. Additional info about answer on research question or answer  about hypothesis testing should be included in the abstract.

  • Introduction and Theory:

Reserarch gap- missing in the text. Should be added. Research gap should be clearly presented in the Introduction chapter.

Reserarch problem- missing in the text. Should be added. Line 37 - correct is cost, time and quality and not cost, schedule and quality. Lines 61 - 74- missing references in the text. Research problem should be clearly presented in the Introduction chapter.

Hypothesis or research questions - are missing before theory chapters.No research questions or hypothesis are presented. This is basic/origin for any research and should be improved in the paper.Due to a lack of research questions or hypothesis after literature overview chapter is difficult to get overall and impression about all aspects of this uncomplete article.

Figure 1- should include Hypothesis or construct scheme .... relations (H1, H2,..etc). Now are missing. Line 354 - authors refer to  hypothesis ?... which hypothesis ??? - there are no hypothesis presented in the paper...???

Hypothesis - Should be added in the text - in the Figure 1 or in the Methodology chapter.

Population and sample- should be better explained in the chapter 3.2. Table 3 and 4 should be explained. Some text, explanations should be after Table 3 and Table 4.

Page 11- Figure 3 - name should be under the figure in the line 390. Pages 13, 14, 25.. there are numerous Figures with no name (?)...Each Figure should have number and name... The same is on pages 16-17...there are numerous Figures with no name (?)...Each Figure should have number and name...

Line 714 - authors write - ''This research provides actionable strategies for enhancing a shared vision, 714 thereby strengthening behaviors and achieving more effective project management out- 715 comes in the construction industry.'' - but where in the article is that ???

Conclusion:

Lines 736 - 752 - Each research question or Hypothesis should be clearly answered in the conclusion.

Reviewer 2 Report

Comments and Suggestions for Authors

The research seeks to understand the interaction between inter-organizational relationship cognition and value addition in engineering projects using SEM and FCM.

The topic is relevant and somewhat original, but lacks clearly articulated research gap which reduces its impact.

The paper attempts to contribute by using a novel methodological combination, but the contribution is diminished by the weak methodological explanation and integration.

The study lacks explicit research questions and hypotheses, making it difficult to follow the analytical framework. The article need to clearly articulate the research questions and hypotheses, provide detailed methodological explanations (The research design and methods, particularly the application of SEM and FCM, are not clearly explained), and consider additional controls such as confounding variables.

The conclusions are somewhat consistent with the results but lack depth and thorough integration with the study’s objectives.

The references are generally appropriate, but a more focused selection could improve the paper.

The paper would benefit from additional tables and figures to better present and clarify the results.

The paper requires significant revisions to improve its methodological rigor, coherence, and overall contribution to the field.

Reviewer 3 Report

Comments and Suggestions for Authors

The paper concerns inter-organizational relationship research. Originality of this paper relies on combining SEM with Fuzzy Cognitive Maps (FCM)

The literature survey requires extension, in my opinion Authors should review other similar research,

There is lack of identification of items, hence I do not know what questions were asked and why

Lack of justification of items.

Authors focus on estimation however, lack of precisely explained latent variables in their theoretical background.

The conclusions are very general. Authors are requested to provide recommendations to project managers, if they locate their research in project management domain.

In section 4.2. figure has no number

Round 2

Reviewer 1 Report

Comments and Suggestions for Authors

According to Editor's decision.

Reviewer 2 Report

Comments and Suggestions for Authors

The authors have made substantial improvements to the article in response to the previous review comments. The added details, red-font revisions, and clarifications have strengthened the paper’s overall quality.

However, some issues remain that need further attention, particularly regarding the explicit statement of research questions, a more critical discussion of results, and the presentation of practical implications.

To be clear:

The research questions are still somewhat implicit. A clearer and more explicit articulation of the research questions or hypotheses would strengthen the paper’s focus.

The conclusion could still be more specific about how the findings contribute to the field. It would benefit from more concrete suggestions for practitioners.